fluff: exploratory analysis and visualization of high-throughput sequencing data

Georgiou Georgios
van Heeringen Simon J. s.vanheeringen@science.ru.nl
Radboud University, Molecular Developmental Biology , Nijmegen , The Netherlands
Campagne Fabien
Electronic publication date: 2016 Jul 19
Publication date: 2016
Volume: 4
Electronic Location ID: e2209
Received 2016 Apr 7; Accepted 2016 Jun 13
Copyright: ©2016 Georgiou and Van Heeringen
Copyright year: 2016
Copyright holder: Georgiou and Van Heeringen
License: This is an open access article distributed under the terms of the Creative Commons Attribution License, which permits unrestricted use, distribution, reproduction and adaptation in any medium and for any purpose provided that it is properly attributed. For attribution, the original author(s), title, publication source (PeerJ) and either DOI or URL of the article must be cited.
License URL: https://creativecommons.org/licenses/by/4.0/

Keywords: ChIP-seq, Clustering, Next-generation sequencing, High-throughput sequencing, Visualization, Python

Funding: The Netherlands Organisation for Scientific Research (NWO-ALW) 863.12.002 US National Institutes of Health (NICHD) R01HD069344 This work was supported by the Netherlands Organisation for Scientific Research (NWO-ALW) [863.12.002 to SJvH]. GG was supported by the US National Institutes of Health (NICHD) [R01HD069344]. The funders had no role in study design, data collection and analysis, decision to publish, or preparation of the manuscript.

==============================
Summary. In this article we describe fluff, a software package that allows for simple exploration, clustering and visualization of high-throughput sequencing data mapped to a reference genome. The package contains three command-line tools to generate publication-quality figures in an uncomplicated manner using sensible defaults. Genome-wide data can be aggregated, clustered and visualized in a heatmap, according to different clustering methods. This includes a predefined setting to identify dynamic clusters between different conditions or developmental stages. Alternatively, clustered data can be visualized in a bandplot. Finally, fluff includes a tool to generate genomic profiles. As command-line tools, the fluff programs can easily be integrated into standard analysis pipelines. The installation is straightforward and documentation is available at http://fluff.readthedocs.org.

Availability. fluff is implemented in Python and runs on Linux. The source code is freely available for download at https://github.com/simonvh/fluff.

Introduction

The advances in sequencing technology and the reduction of costs have led to a rapid increase of High-Throughput Sequencing (HTS) data. Applications include chromatin immunoprecipitation followed by high-throughput deep sequencing (ChIP-seq; Robertson et al., 2007) to determine the genomic location of DNA-associated proteins, chromatin accessibility assays (Buenrostro et al., 2013; Hesselberth et al., 2009) and bisulfite sequencing to assay DNA methylation (Lister et al., 2009). The integration of these diverse data allow identification of the epigenomic state, for instance in different tissues (Martens & Stunnenberg, 2013; Roadmap Epigenomics Consortium et al., 2015) or during development (Hontelez et al., 2015). However, the scale and complexity of these datasets call for the use of computational methods that facilitate data exploration and visualization.

Various options exist to explore and visualize HTS data mapped to a reference genome, for instance in aggregated form such as heatmaps and average profiles. These include general purpose modules for specific programming languages (Huber et al., 2015), dedicated HTS modules (Dale, Matzat & Lei, 2014; Statham et al., 2010; Akalin et al., 2015), command-line tools (Shen et al., 2014; Giannopoulou & Elemento, 2011), web tools (Ramírez et al., 2014), stand-alone applications (Ramírez et al., 2014; Ye et al., 2011) and tools that depend on other software for visualization (Heinz et al., 2010). Here, we present fluff, a Python package for visual, reference-based HTS data exploration. It includes command-line applications to both cluster and visualize aggregated signals in genomic regions, as well as to create genome browser-like profiles. The scripts can be included in analysis pipelines and accept commonly used file formats. The fluff applications are pitched at the beginner to intermediate user. They have sensible defaults, yet allow for customizable creation of high-quality, publication-ready figures.

Methods

General

Detailed documentation, including tutorials, is available at http://fluff.readthedocs.org. Fluff is implemented in Python, and uses several previously published modules (Brewer, 2016; Anders, Pyl & Huber, 2015; Dale, Pedersen & Quinlan, 2011; Quinlan & Hall, 2010; Li et al., 2009; De Hoon et al., 2004, see Supplemental Information). All fluff tools support indexed BAM, bigWig or (tabix-indexed) BED, WIG or bedGraph files as input. A large selection of major image formats are supported as output. The fluff tools were developed to explore ChIP-seq data, however, they will work with any type of data where (spliced) reads can be mapped to a genomic reference. For instance DNA methylation profiles from bisulfite-sequencing or RNA-seq data (Fig. S1) can also be visualized.

Normalization

Normalization of sequencing data is critical for downstream analysis and various methods have been proposed (see for instance Angelini et al., 2015 and Bailey et al., 2013 for an overview of ChIP-seq normalization methods). For visualization, the most important factor is the sequencing read depth. Therefore fluff has the option to normalize to the total number of mapped reads. Alternatively, averaged signal files such as bigWig tracks that are processed or normalized by a different method can be used as input.

Program descriptions

Heatmaps

Visualization of HTS data as heatmaps, where rows represent different genomic regions, can highlight important aspects of the data, like differential enrichment or positional patterns for specific groups of features. In addition, it allows for comparison between multiple regions within the same or between different experiments. The fluff heatmap tool visualizes HTS data on basis of a list of genomic coordinates. The data can optionally be clustered using either k-means or hierarchical clustering. For clustering, the read counts in the bins are normalized to the 75 percentile. The distance can be calculated using either the Euclidean distance or Pearson correlation similarity.

If the regions in the input file are not strand-specific, different clusters might represent the same strand-specific profile in two different orientations. Clusters that are mirrored relative to the center can optionally be merged. Here, the similarity is based on the chi-squared p-value of the mean profile per cluster.

One important use case for clustering is the ability to identify dynamic patterns, for instance during different time points or conditions. For this purpose, clustering on the binned signal is not ideal. Therefore, fluff heatmap provides the option to cluster genomic regions based on a single value derived from the number of reads in the feature centers (+∕ − 1 kb). In combination with the Pearson correlation metric, this allows for efficient retrieval of dynamic clusters. The difference is illustrated in Fig. 2.

Bandplots

In heatmaps, more subtle patterns can be difficult to detect, as the dynamic range of signal intensities is not well-reflected in the color scale. Therefore, as an alternative to a heatmap, fluff bandplot plots the average profiles in small multiples (Shoresh & Wong, 2012). Here, the spatial encoding of the signal allows for more accurate comparison of values (Gehlenborg, Nils & Bang, 2012). The median enrichment is visualized as a black line with the 50th and 90th percentile as a dark and light colour respectively.

Profiles

Genome browsers are unrivaled for data exploration and visualization in a genomic context. However, it can be useful to create profiles of HTS data in genomic intervals using a consistent command-line tool, that can optionally be automated. The fluff profile tool can plot summarized profiles from one or more profiles, together with (gene) annotation from a BED12-formatted file.

Analysis

In short, FASTQ files were download from NCBI GEO (Edgar, Domrachev & Lash, 2002) and mapped to the human genome (hg19) using bwa (Li & Durbin, 2009). Duplicate reads were marked using bamUtil (http://genome.sph.umich.edu/wiki/BamUtil). All BAM files from replicate experiments were merged. Peaks were called using MACS2 (Zhang et al., 2008) with default settings. See the Supplemental Information for specific details and accession numbers.

Results

Demonstrating fluff: dynamic enhancers during macrophage differentiation

To illustrate the functionality of fluff we visualized previously published ChIP-seq data (Saeed et al., 2014). Here, the epigenomes of human monocytes and in vitro-differentiated naïve, tolerized, and trained macrophages were analyzed, with the aim to understand the epigenetic basis of innate immunity. Circulating monocytes (Mo) were differentiated into three macrophages states: to macrophages (Mf), to long-term tolerant cells (LPS-Mf) by exposition to lipopolysaccharide and to trained immune cells (BG-Mf) by priming with β-glucan. We used fluff heatmap to cluster and visualize the signal of histone 3 lysine 27 acetylation (H3K27ac), which is located at active enhancers and promoters (Fig. 1A). The input consisted of a BED file with 7,611 differentially regulated enhancers (Table S1) and four BAM files, for each of the monocytes and three types of macrophages. Using k-means clustering (k = 5) with the Pearson correlation metric, the heatmap recapitulates the H3K27ac dynamics as described (Saeed et al., 2014).

Figure 1 An example of the fluff output.

All panels were generated by the fluff command-line tools and were not post-processed or edited. (A) Heatmap showing the results of k-means clustering (k = 5, metric = Pearson) of dynamic H3K27ac regions in monocytes (Mo), naïve macrophages (Mf), tolerized (LPS-Mf) and trained cells (BG-Mf) (Saeed et al., 2014). ChIP-seq read counts are visualized in 100-bp bins in 24-kb regions. (B) Bandplot showing the average profile (median: black, 50%: dark color, 90%: light color) of the clusters as identified in Fig. 1A. (C) The H3K27ac ChIP-seq profiles at the CNRIP1 gene locus, which shows a gain of H3K27ac in Mf, LPS-Mf and BG-Mf relative to Mo.

Figure 2 Example of the output of fluff heatmap using standard clustering compared to using the dynamics option.

Shown are the H3K27ac ChIP-seq read counts in 100 bp bins in 20 kb around the DNaseI peak summit in human H1 ES cell-derived cells. (A) Heatmap showing the results of k-means clustering of all bins (k = 7, metric = Euclidean) (B) Heatmap showing the results of k-means clustering in 2 kb regions centered at the peak summit (k = 7, metric = Pearson).

While heatmaps are often used for visualization of signals over genomic features, either clustered or ordered by signal intensity, it can be difficult to distinguish relative levels of individual clusters. Figure 1B shows an alternative visualization of average enrichment profiles in small multiples. The same clusters as in Fig. 1A are plotted using fluff bandplot. Shown are the median (black line), along with the 50th (darker color) and 90th percentile (lighter color) of the data. This allows for more detailed comparisons.

Finally, we illustrate fluff profile, which can visualize one or more genomic regions (Fig. 1C). This figure highlights the CNRIP1 gene from cluster 2, which shows a consistent increase of H3K27ac from Mo to Mf, LPS-Mf and BG-Mf. The signal profiles are directly generated from the BAM files.

Identification and visualization of dynamic patterns

Most applications that cluster HTS data for heatmap visualization use a binning approach, followed by clustering using the Euclidean distance. The implicit effect is that the bins are clustered on basis of the spatial patterns relative to the region of interest. Often, this is the desired result, for instance when clustering the ChIP-seq enrichment patterns of different histone modifications at the transcription start sites of genes. However, for other analyses this clustering approach does not suffice. An example could be the ChIP-seq profiles of specific histone modifications correlated to the activity of a regulatory element, such as H3K4me3 at promoters or H3K27ac at enhancers. In this case, a relevant objective is to identify the clusters associated with differential activation dynamics. As illustration, we visualized the H3K27ac enrichment profile at DNaseI hypersensitive sites in human embryonic stem (ES) cells differentiated into different lineages (Xie et al., 2013). Here, H1 ES cells were differentiated into mesendoderm, neural progenitor cells, trophoblast-like cells, and mesenchymal stem cells. We first clustered the H3K27ac profiles at regulatory elements on chromosome 1 using the standard approach, based on comparing all the bins using the Euclidean distance metric (Fig. 2A).

Here, we identify two clusters with high enrichment (cluster 3 and cluster 5), a cluster with relatively low, narrow enrichment (cluster 1), and two clusters with broad enhancer domains (cluster 4 and 6). However, only two strong dynamic clusters are identified, cluster 2, which shows enhancers specifically activated in mesenchymal stem cells and cluster 7 which shows enhancers specifically activated in trophoblast-like stem cells. Figure 2B shows an alternative clustering approach implemented in fluff heatmap. Here the regions were clustered on basis of the Pearson correlation of read counts in the center of the region (extended to 2 kb). This shows a completely different picture and we now can identify enhancers specific to H1 ES cells (cluster 5), mesenchymal (cluster 4), mesendoderm (cluster 7), neuronal progenitor (cluster 3) and trophoblast cells (cluster 6). These lineage-specific enhancer dynamics were not visible in the clustering in Fig. 2A.

Conclusion

The analysis of multi-dimensional genomic data requires methods for data exploration and visualization. We provide fluff, a Python package that contains several command-line tools to generate figures for use in high-throughput sequencing analysis workflows. We aim to fill the gap between powerful, flexible libraries that require programming skills on the one hand, and intuitive, graphical programs with limited customization possibilities on the other hand. These tools were developed based on a need for straightforward analysis and visualization of ChIP-seq data and have been successfully applied in a variety of projects (Menafra et al., 2014; Van den Boom et al., 2016; Kouwenhoven et al., 2015). In conclusion, fluff helps to interpret genome-wide experiments by efficient visualization of sequencing data.

Supplemental Information

Table S1 List of dynamic enhancers

Click here for additional data file.

Supplemental Information 1 Supplemental Information

Click here for additional data file.

This study makes use of data generated by the Blueprint Consortium. A full list of the investigators who contributed to the generation of the data is available from www.blueprint-epigenome.eu. Additionally, this study used data provided by the NIH Roadmap Epigenomics Consortium (http://nihroadmap.nih.gov/epigenomics/).

Additional Information and Declarations

Competing Interests

Author Contributions

Data Availability

The authors declare there are no competing interests.

Georgios Georgiou conceived and designed the experiments, performed the experiments, analyzed the data, contributed reagents/materials/analysis tools, wrote the paper, prepared figures and/or tables, reviewed drafts of the paper, wrote the code.

Simon J. van Heeringen conceived and designed the experiments, contributed reagents/materials/analysis tools, wrote the paper, reviewed drafts of the paper, wrote the code.

The following information was supplied regarding data availability:

Code DOI: 10.5281/zenodo.54949

Github: https://github.com/simonvh/fluff.

Example data DOI: 10.6084/m9.figshare.3113728.v1

Figshare: https://figshare.com/articles/fluff_example_data/3113728.

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
