# Peer review of "fluff: exploratory analysis and visualization of high-throughput sequencing data"

_PeerJ, doi:10.7717/peerj.2209_

## Round 0.1 · original submission · Major Revisions

I agree with the comments of the reviewers and suggest to carefully revise the manuscript following their suggestions.

Reviewer 1 ·

Basic reporting

The paper is well written, with minor typos/errors (see below). Figures are relevant, well designed and labeled. Test data are provided and it is easy for the reader to follow the website instructions to recreate the figures. One suggestion to the authors is to mention that the fluff package works with Python 2.7 (Python 3.7 gave me inconsistencies with other packages).

The authors state that fluff can be used to visualize High-Throughput Sequencing data. However, they show examples of visualizing only ChIP-seq, open chromatin sites and DNA methylation data, which is not the whole spectrum of High-Throughput Sequencing. Can these types of visualization be applied to other HTS datasets, such as RNA-seq and exome-seq, two of the most widely used and very popular sequencing applications? I suggest the authors either show the applicability of fluff in these seq technologies, or state the reasons fluff could not be used for RNA-seq/exome-seq. If the former is true, Abstract and Introduction sections have to be rewritten to reflect the application of fluff to particular sequencing technologies.

Also, related work is much broader than the references provided by the authors. There are several popular analysis tools (both command-line and R-based), specific to ChIP-seq and methylation seq datasets that do provide these types of visualization, like Homer, ChIPseeqer, and methylkit. I suggest that the authors compare fluff with the visualization options provided by these tools.

Typos/Errors
Line 48: analystics → analytics
Line 144: which visualizes one more genomic regions → rephrase

Experimental design

One concern is that BAM files are used to create the visualizations, while it is known that normalized reads should be used instead in order to compare different samples (e.g., normalize by the total number of mapped reads). This is a major concern since different number of total reads per sample can result in confusing results visually in the heatmaps, or in density tracks. How do the authors calculating ChIP-seq read densities across different experiments? Also, can .wig/.bigwig files be used instead of BAM files?

Validity of the findings

The results look valid (I was able to produce the figures myself). However, creating heatmaps/bandplots and profiles is not a novel idea for Next Generation Sequencing datasets. This paper looks more like a technical report or a tutorial, rather than a research paper. The main advantage I find for this package is its integration into established pipelines. I suggest that the users point this out more.

Additional comments

This paper describes a Python package with visualization functionalities for ChIP-seq, chromatin accessibility assays (DNase-seq), and DNA methylation data.
• Creating heatmaps/bandplots and profiles is not a novel idea for Next Generation Sequencing datasets. This paper looks more like a technical report or a tutorial, rather than a research paper. The main advantage I find for this package is its integration into established pipelines. I suggest that the users point this out more.
• The authors state that fluff can be used to visualize High-Throughput Sequencing data. However, the examples mentioned/shown are only for ChIP-seq, open chromatin sites and DNA methylation data, which is not the whole spectrum of High-Throughput Sequencing. Can these types of visualization be applied to other HTS datasets, such as RNA-seq and exome-seq, two of the most widely used and very popular sequencing applications? I suggest the authors either show the applicability of fluff in these seq technologies, or state the reasons fluff could not be used for RNA-seq/exome-seq. If the former is true, Abstract and Introduction sections have to be rewritten to reflect the application of fluff to particular sequencing technologies.
• Expand related work popular analysis tools specific to ChIP-seq and methylation seq datasets like Homer, ChIPseeqer, and methylkit. I suggest that the authors compare fluff with the visualization options provided by these tools.
• One concern is that BAM files are used to create the visualizations, while it is known that normalized reads should be used instead in order to compare different samples (e.g., normalize by the total number of mapped reads). This is a major concern since different number of total reads per sample can result in confusing results visually in the heatmaps, or in density tracks. How do the authors calculating ChIP-seq read densities across different experiments? Also, can .wig/.bigwig files be used instead of BAM files?
• One suggestion to the authors is to mention that the fluff package works with Python 2.7 (Python 3.7 gave me inconsistencies with other packages).
• Line 48: analystics → analytics
• Line 144: which visualizes one more genomic regions → rephrase

·

Basic reporting

Not applicable

Experimental design

Not applicable

Validity of the findings

Not applicable

Additional comments

The paper describes a set of command-line tools for the visualization of sequencing data. Specifically, a heat map and a band plot can be produced, and various clustering algorithms/distance measures can be used.

The Methods section is unfortunately not very focused: the Implementation section lists packages/libraries that were used in the tool and gives installation information. A very brief mention at the end of the paper of these things might be relevant, but generally this information should go into developer/user documentation.

The program description section partially reads like a user manual but also contains some useful information. However, a scientific article is distinct from a manual in that it details the justifications of the design. Yet such an explanation is missing. Why was a heatmap and a bandplot chosen? Why are the read counts normalized? [There are many more examples] These aren't necessarily bad choices, but I don't see any arguments for them.


The data analysis section goes into significant detail on data sources and preparations. While this information is critical when demonstrating biological findings, it is of minor importance to a discussion of the tool and should be relegated to the supplementary material (if at all). The same is true for Table 1, 2 and the commands / parameters used: readers that want to evaluate whether the tool is relevant for their usecase do not care about this: only when a user wants to use the tool does this become relevant, and hence it should be in the documentation, not in the paper.

The Results section is well done - it contains an appropriate level of detail and gives some of the design justification that is missing from the Methods section.


I generally believe that the tools, while not revolutionary, can address a clear need in the scientific community and are largely well executed. The paper, however, needs a significant revision to make it relevant to readers that wants to evaluate the fitness for use for their tasks, and shouldn't be inflated with details about implementation and usage.

---

## Round 0.2 · accepted · Accept

I believe that your revised manuscript has addressed the previous comments of the reviewers. The manuscript reads well and should be of interest to readers interested in visualization of ChipSeq data.